# The Burden of Fungal Infections in Ethiopia

**DOI:** 10.3390/jof5040109

**Published:** 2019-11-22

**Authors:** Tafese B. Tufa, David W. Denning

**Affiliations:** 1Asella Teaching and Referral Hospital, College of Health Sciences, Arsi University, P.O. Box 04, Asella, Ethiopia; 2Hirsch Institute of Tropical Medicine (HITM), Heinrich-Heine University, P.O. Box 04, Asella, Ethiopia; 3The National Aspergillosis Centre, Wythenshawe Hospital, Manchester M23 9LT, UK; 4The University of Manchester and Manchester Academic Health Science Centre, Manchester M13 9PL, UK; ddenning@manchester.ac.uk

**Keywords:** invasive fungal infections, tinea capitis, epidemiology, Ethiopia

## Abstract

The burden of severe fungal infections (FIs) is not well addressed in Ethiopia. We have estimated the burden of FIs from multiple demographic sources and by searching articles from PubMed. Opportunistic FIs were estimated using modelling and 2017 national HIV data. The burdens of chronic pulmonary aspergillosis (CPA) and allergic bronchopulmonary aspergillosis (ABPA) were estimated by using the prevalence of asthma, chronic obstructive pulmonary disease, and annual the incidence of tuberculosis. Of the 105,000,000 estimated Ethiopian population, 610,000 are thought to have HIV infection. Our estimation of HIV-related FIs were: 9900 cryptococcal meningitis (CM), 12,700 *Pneumocystis jirovecii* pneumonia (PCP), 76,300 oral and 56,000 oesophageal candidiasis cases. A remarkable 7,051,700 4–14-year-olds probably have tinea capitis and 1,469,000 women probably have recurrent *Candida* vaginitis. There were 15,200 estimated CPA cases (prevalence) and 11,500 invasive aspergillosis (IA) cases (incidence). Data are scant, but we estimated 5300 candidaemia and 800 Candida peritonitis cases. In conclusion, approximately 8% of Ethiopians suffer from FIs annually, mostly schoolchildren with tinea capitis. IA, CM and PCP are the major causes of fungal deaths. The absence of CD4 count is challenging the identification of HIV patients at risk of opportunistic FIs. There is a pressing need to improve FI diagnosis, probably including national surveillance.

## 1. Introduction

Pathogenic, opportunistic and allergenic fungi cause a very wide range of diseases from simple superficial mycosis to complex disseminated endemic or opportunistic mycosis [1]. Unlike superficial or cutaneous mycosis, disseminated fungal infections (FIs) are life threatening if not appropriately treated. In resource-limited settings (RLSs), invasive FIs remain understudied and underdiagnosed despite their high mortality rates when compared with other infectious diseases [1,2].

The number of immunosuppressed individuals who are at risk of developing opportunistic mycosis is increasing due to HIV/AIDS even though the rate is reduced because of antiretroviral therapy (ART), the expansion of intensive care units (ICUs), chronic diseases, malignancies and sophisticated procedures such as stem cell or organ transplantation [3]. The most common opportunistic FIs are cryptococcal meningitis (CM), Pneumocystis jirovecii pneumonia (PCP), invasive candidiasis and disseminated aspergillosis [4]. Diagnosis and management of these opportunistic mycosis in RLSs is challenging as standard diagnostic methods and essential antifungal medicines are often absent [5].

Many populations are sensitive (allergic) to fungi and develop severe asthma or exacerbation of pre-existing respiratory diseases [6]. Anyone in the general population may develop dermatophytoses and vaginal candidiasis is common in women [1]. Although less attention is given to the management of dermatophytoses, the prevalence of atypical dermatophytoses is increasing and has become an emerging public health problem, with the new development of extensive terbinafine resistance [7]. Antifungal resistance in Trichophyton spp. is due to two amino acid substitutions or point mutations [8]. As studies from India indicate, the predominant causative organism of dermatophytoses has shifted from T. rubrum to T. mentagrophytes [9,10,11]. The two main causative agents of dermatophytoses survive on towels for <12 weeks for T. rubrum but >25 weeks for T. mentagrophytes [12]. This change may be responsible for the widespread and inflammatory lesions associated with T. mentagrophytes and transmitted by sharing fomites.

Across the world, over one billion people of all ages suffer from FIs annually [13], accounting for over 1.6 million deaths and contributing to the poor and fatal outcome of many more diseases [14]. This figure is three-fold higher than the annual deaths from malaria and comparable to the number of tuberculosis deaths per year [2].

Even though invasive FIs are life threatening, they do not receive sufficient attention from many global health organizations or governments. Particularly in sub-Saharan African countries, data on the incidence and prevalence of these infections are lacking and limited diagnostic facilities, few evidence-based case management protocols and the unavailability of appropriate antifungal medications and specialists in the field amplify the problems [1,2]. This situation results in high and preventable case mortality and morbidity.

Robust estimates of fungal disease incidence and prevalence are lacking in Ethiopia [2]. In the absence of either population- or hospital-based surveys as source epidemiological data, we sought to provide estimates by modelling using available data on FIs in Ethiopia and assumptions drawn from published literature. This estimation of FIs in Ethiopia will allow a baseline estimate for future studies and serve as a reference for policy makers.

## 2. Materials and Methods 

### Data Sources

For the estimation of the serious FI burden in Ethiopia, we used the 2017 estimated Ethiopian total population for the general healthy population. We used different risk groups including people living with HIV/AIDS (PLWHA), patients with asthma, chronic obstructive pulmonary disease (COPD), cancer, pulmonary tuberculosis and postsurgical patients, and those admitted to ICUs to estimate specific infections. To estimate tinea capitis prevalence, we determined the number of children in the high-risk age group (3–14 years old) from the Ethiopia National Education [15] and Education for All National Review Report [16]. Data on PLWHA numbers were obtained from the HIV/AIDS [17] 2017 Global report [18]. Data about tuberculosis annual incidence in Ethiopia were obtained from the 2017 World Health Organization (WHO) tuberculosis report [19]. We also used data from published reports to identify the most accurate numerators and denominators for each FI as described below.

We searched published articles from PubMed related to FIs in Ethiopia and used the prevalence and/or incidence rates of FIs for our estimations. Where published data was missing, we used unpublished data from government institutions of regional or national reports. In the absence of local data from the country, published data were used from neighbouring countries. In our modelling, different populations are included as risk groups for particular FIs as risks vary. Therefore, systematic reviews and meta-analysis were not applicable to compare the heterogeneity. A narrative approach was taken to report the findings of the studies included. Our estimates are based only on well-defined risk populations, which might be the lowest incidence rates of FIs and rounded to the nearest 100 to be less precise. 

## 3. Results

### 3.1. Country Profile

In 2017, the Ethiopian total population was estimated to be 105 million, with 51.3% being female (Table 1). Of the general population, 41% were younger than 15 years and only 5% of Ethiopians are over age 60 years [20]. The gross domestic product per capita was USD 768 in 2017. According to the WHO 2017 report, 610,000 Ethiopians were estimated to be HIV-infected, with 170,000 of them not receiving antiretroviral (ARVs), and of those on ARVs, less than 50% had HIV viral load suppression [17,21]. New AIDS cases and AIDS-related deaths have been decreasing sharply since 2011, with a fall from 32,000 to 15,000 deaths in 2017 [17,21]. Of all PLWHA in 2017, 50% were receiving ART. Approximately 184,500 adult HIV-positive patients had CD4 counts <200 cells/µL and we estimated that 50% of them were severely immunosuppressed and therefore at a higher risk of opportunistic FIs (Table 1).

### 3.2. Cryptococcal Meningitis

CM is among the leading causes of mortality among adult HIV/AIDS patients in sub-Saharan Africa [42,43]. Early screening with cryptococcal antigen (CrAg) among the target population or advanced HIV-infected patients and giving pre-emptive treatment can prevent the development of CM and associated mortality. The prevalence of cryptococcal antigenaemia ranges from 6% to 11.7% among low CD4 T-cell count or among HIV/AIDS patients [25,26,27,28,29] (Table 2). We estimated the annual burden of CM by applying the 11.7% rate of cryptococcal antigenaemia among the 169,500 HIV patients thought to be at risk and calculated 9900 cases of cryptococcosis (9.4 cases per 100,000 person-years) (Table 3). If left untreated, all cryptococcal antigenaemia patients will develop CM [44]. We have not attempted to estimate non-HIV-related cases of CM. 

### 3.3. Pneumocystis jirovecii Pneumonia

PCP is among the more common life-threatening opportunistic mycoses and may be mistaken as bacterial pneumonia due to a lack of standard diagnosis and management (including high-dose corticosteroids) in RLSs [45]. Early rapid diagnosis and treatment drive survival has outcomes of >70% in Africa and up to 90% in developed countries [45].

In Ethiopia, all patients with advanced immunosuppression (CD4 count less than 200 cells/µL) due to HIV are recommended to receive co-trimoxazole prophylaxis for prevention of PCP. The maximum prevalence of PCP diagnosed by nested PCR was 30% irrespective of ART status or co-trimoxazole prophylaxis [46] (Table 2). We used 7.5% as the minimum prevalence (or ¼ of the maximum prevalence) to avoid over the estimation of PCP cases due to oversensitive nested PCR or *Pneumocystis jirovecii* colonization (Table 2). 

We therefore estimated an incidence of 12,700 cases of PCP in 2017 (12.1 cases per 100,000 person-years) assuming that 15% of all 169,500 adults are at risk over 2 years (Table 3). Due to lack of data, we could not make a reliable estimate of the incidence of PCP in children among those infected with HIV, or infections in non-HIV-infected adults.

### 3.4. Oral Candidiasis

Approximately 50% newly HIV diagnosed patients with low CD4 T-cell count commonly show oral candidiasis as an initial presentation of symptomatic HIV-infected patients [47,48]. Over the course of a year, this rate rises to ~90% of PLWHA not taking ART [32,49] (Table 2).

We estimated oral candidiasis to affect 76,300 Ethiopian at a rate of 72.6 per 100,000 person-years (Table 3). This burden was estimated by assuming that 90% of the patients at risk with advanced HIV (169,500 cases) will develop oral candidiasis over 2 years (Table 1).

### 3.5. Oesophageal Candidiasis

Oesophageal candidiasis is a common AIDS-defining illness occurring in those with advanced HIV, thus commonly among ART-naïve patients or ART-defaulters. 

Oesophageal candidiasis cases were estimated to be 55,900 at a rate of 53.3 per 100,000 person-years (Table 3) under the assumptions that 20% of AIDS patients at risk, and 0.5% of those on ART develop oesophageal candidiasis [33] (Table 2).

### 3.6. Allergic Fungal Respiratory Diseases

Allergic bronchopulmonary aspergillosis (ABPA) is an occasional complication of asthma and cystic fibrosis (CF). Chronic *Aspergillus fumigatus* airways infection is a feature of chronic pulmonary aspergillosis (CPA) [50,51]. One study from South Africa reported a 2.5% prevalence of ABPA among adults with asthma presenting to secondary care [34] (Table 2).

We used data collected as part of the World Heath Survey (2002–2003) for Ethiopia on the prevalence of adult asthma (2.0%) to estimate the prevalence of fungal asthma in adults in Ethiopia (23) (Table 1). There are no reliable published data for CF prevalence.

Assuming a 2% adult asthma prevalence [23] (1,203,600 adults) (Table 1) and the estimated 2.5% ABPA prevalence among asthmatic adults (Table 2) [34], 30,100 adults were estimated to have ABPA in 2011 (Table 3). We only used asthma prevalence among adults, as asthma is more common in children and ABPA is less common. [34,52]. CPA is exclusively an adult disease.

We also estimated the burden of severe asthma with fungal sensitisation (SAFS) in Ethiopia from the adult asthmatic population. Among these adult asthmatics, approximately 10% will suffer from severe asthma. Of these, 33% have been reported to be sensitized to *Aspergillus* or another mould [34] (Table 2). We calculated from these assumptions that SAFS affects 40,900 Ethiopians—a prevalence of 38.9 cases per 100,000 person-years (Table 3).

### 3.7. Chronic Pulmonary Aspergillosis (CPA)

CPA is a long-term sequel of several lung conditions, often following treated pulmonary tuberculosis [51,53]. In RLSs and in tuberculosis-endemic areas such as Ethiopia, a lack of diagnosis of *A. fumigatus* causing CPA probably leads to underdiagnoses and mismanagement. Such patients are mostly treated as cases of sputum smear-negative (and Xpert® negative) tuberculosis [54]. CPA presents with several different radiological features including simple aspergilloma, chronic cavitary pulmonary aspergillosis and chronic fibrosing pulmonary aspergillosis [53]. 

We calculated the annual incidence and prevalence of CPA using tuberculosis annual incidence and by applying the model described by Denning et al. from 2011 [53] (Table 2). In 2016, tuberculosis was reported in 182,000 individuals—of which, it is assumed that 68% are pulmonary and 85% survived, leaving 105,200 [22] (Table 1). According to Page et al. in a prospective study in Uganda, the annual rate of CPA was 6.5% among those with a cavity in the 2–7 years after completing therapy for pulmonary tuberculosis and 0.2% in those without visible cavities [35] (Table 2). This translates to 2200 cases annually and, assuming 10% mortality, the CPA prevalence after TB is estimated at 8100 cases. This estimate omits two other elements: those who were misdiagnosed as having TB and in fact had CPA and did not survive long enough to be assessed in the Uganda study and those with other underlying diseases such as COPD, pneumothorax and other pulmonary disorders. We have assumed that post-TB CPA is 67% of the total and that pulmonary tuberculosis is responsible for 80% of all CPA cases in Ethiopia. Overall, we estimated 15,200 cases of CPA at a rate of 14.5 cases per 100,000 persons per years (Table 3).

### 3.8. Invasive Aspergillosis (IA)

We estimated 700 cases of IA in 2011—an annual incidence of 0.65 per 100,000 person-years. We calculated this from the most immunosuppressed patients with leukaemia, assuming that 10% of patients with acute myeloid leukaemia will develop IA [53,55], and that there is an equal number of people with IA complicating all other leukaemias and lymphoma patients. We also assumed that 2.6% of lung cancer patients will develop IA [56] and that 4% of AIDS deaths are attributable to IA [44]. There are an estimated 7.8 people over 40 years of age with COPD and approximately 10% are admitted to hospital each year. Assuming 1.3% develop IA [57], we anticipate an annual IA incidence of 10,200 in COPD patients. Overall, we estimated 11,500 IA cases at a rate of 10.92 cases per 100,000 person-years (Table 3). 

### 3.9. Candida Infections

Recurrent vulvovaginal candidiasis (rVVC) is defined as four or more episodes per year [36]. rVVC is usually caused by *Candida albicans*. *Candida glabrata*, which is fluconazole resistant, and other species are implicated less often. The absence of rapid, simple, and inexpensive diagnostic tests continues to result in both over- and underdiagnoses of vulvovaginal candidiasis at the same time. It is thought that approximately 70% of women suffer from vulvovaginal candidiasis at least once in their lives, more often during pregnancy [36].

Recurrent attacks of vulvovaginal candidiasis have been estimated to affect 5–10% of women annually [58,59], primarily based on data from Foxman et al [36,37] (Table 2). Estimates of rVVC are intrinsically difficult to conduct to reliably estimate the incidence and prevalence of rVVC among women [59].There are no reports of rVVC from East Africa, but a study in Tunisia found a similar prevalence of 6.1% [60].

We estimated rVVC among adult women aged 15–50 years, assuming that 6% are affected. Recurrent VVC is not common in post-menopausal women in the absence of hormone replacement therapy. Out of a total of 24,482,600 women aged 15–50 years, approximately 1,469,100 women could be affected by rVVC in 2017 at a rate of 1399 women per 100,000 person-years (Table 3). We made no allowance for HIV infection.

The incidence of candidaemia and candida peritonitis were estimated by considering cancer patients, postsurgical patients and patients admitted to ICU wards as high risk for candidaemia case development (Table 1). We estimated that candidaemia affects 5300 women, with an annual incidence of 5 per 100,000 person-year, and that there are 800 episodes of *Candida* peritonitis (intra-abdominal candidiasis), with 0.75 per 100,000 person-years, respectively (Table 3). We calculated the annual incidence of *Candida* peritonitis using the assumption that if candidaemia occurs at a population rate of 5 cases/100,000, one-third are in ICU wards or an equivalent ward and so there will be one patient with *Candida* peritonitis for every two patients with candidaemia in ICUs [53]. It is not easy to confirm this estimate of candidaemia in Ethiopia, as blood cultures are infrequently performed even though we have been observing cases during daily clinical practice. 

### 3.10. Other Fungal Infections

A recent publication indicates that tinea capitis affects 32.3% of primary school children in Ethiopia [38] (Table 2). According to the Education for All 2015 report, there are 21,832,000 nursery school- and primary school-aged children in Ethiopia [16] (Table 1). We estimated from these reports that 7,051,700 school children suffer from tinea capitis each year at a rate of 6716 per 100,000 person-years (Table 3). 

There was only one African histoplasmosis case reported from Ethiopia [61]. 

### 3.11. Estimate of Fungal Infections-Associated Mortality in Ethiopia

Opportunistic invasive FIs such as cryptococcosis, IA and PCP are the principal cause of mortality among people living with HIV in Ethiopia. According to our estimations, they contribute to 20,500 to 34,600 Ethiopian deaths annually. In Ethiopia, CM has been reported to have a 100% fatality rate within a year if not treated with an appropriate antifungal and up to a 68% case fatality rate when treated with high-dose fluconazole monotherapy [25,43]. In 2017, we estimated that out of the total 9900 CM cases, the minimum and maximum CM case mortality would be between 6700 and 9900 (Figure 1).

Available evidence data demonstrate that IA has a case fatality of 90% in leukaemia patients if diagnosed late [62] and if left untreated, it has a case fatality of 100% [63]. We estimated AI could contribute to 10,300 to 11,500 deaths annually in the country.

The case fatality rate of PCP ranges from 16% to 68% in Africa [45]. We used this case fatality to estimate PCP-related annual death in Ethiopia. Out of the total 12,700 PCP cases estimated in Ethiopia, between 2000 and 8600 deaths could occur annually in the country. Denning et al. (2013) reported that the annual CPA fatality rate ranges from 10% to 30% [34] and we calculated that CPA is responsible for 1500 to 4600 deaths annually. In Ethiopia, AIDS remains the leading cause of premature death and years of life lost. Both CM and PCP make major contributions to these many early deaths.

We estimated that over 3600 candidaemia [64] and *Candida* peritonitis patients probability died in the country by using a 60% candidaemia case fatality rate from a report in South Africa [61]. Tinea capitis, ABPA, and recurrent vaginitis are not common causes of death. According to the report in 2013, approximately half a million people died due to severe asthma in the world [65]. However, according to the recent estimated data, the prevalence of fungal asthma was estimated to be 4% and could contribute to 62,544 deaths in Africa alone [66], but it is unclear how many deaths occur in Ethiopia. 

## 4. Discussion

This is the first attempt to estimate the burden of FIs in Ethiopia. Overall, approximately 8% of Ethiopians suffer from serious FIs yearly, including the non-lethal but highly problematic tinea capitis and rVVC. More data are available on the HIV population and tinea capitis than on other groups of patients in Ethiopia. HIV fungal opportunistic infections contribute to 2% of the total FIs and significantly to the number of deaths due to FIs in the country.

IA, CM, PCP, and CPA are the major causes of mycoses-related deaths, though to contribute 33%, 29%, 25%, and 13% of FI deaths, respectively. This is similar to mycoses-related mortality in East African countries [67,68]. Our estimates indicate that most FIs in the country affect PLWHA. This could be because of a host immunity defect due to HIV infections and more data are available on the HIV population in Ethiopia than on other groups of patients with immunosuppressive conditions.

The current Ethiopian national guidelines recommend ART initiation irrespective of CD4 T-cell count following the latest WHO “treat all” guidelines [69]. Currently in the country, there are no CD4 T-cell count test services available in governmental hospitals. Thus, the identification of target populations at greatest risk for opportunistic infections in need of co-trimoxazole and fluconazole prophylaxis is often not possible. The other big problem is the absence of CD4 T-cell testing services in the country, and so screening for cryptococcal infection among HIV advanced patients is not performed. The WHO recommends CrAg screening in HIV-infected patients with a low CD4 T-cell count (<100 cells/µL) prior to ART initiation to prevent CM development and cryptococcal infections associated with immune reconstitution inflammatory syndrome [51]. Delayed ART initiation by 4–6 weeks for CM patients has survival benefits [70]. In a country such as Ethiopia, where CM and PCP are highly prevalent, the viral load alone is not enough to monitor HIV advanced patients.

We estimated 12,700 annual cases of PCP from a study which diagnosed PCP using bronchoalveolar lavage and the nested PCR method among HIV-positive patients with suspected pulmonary tuberculosis. In our study, almost half of the patients were on ART and co-trimoxazole. So, our estimation may be higher than expected as our denominator did not include patients on co-trimoxazole prophylaxis. 

In Ethiopian HIV clinics, PCP is diagnosed clinically as the presence of cough and low oxygen saturation. Even though Toluidine Blue O and Giemsa staining have an acceptable sensitivity and high specificity both in expectorated sputum and bronchoalveolar lavage (BAL) samples for the diagnosis of PCP, these recommended diagnostic methods using induced sputum and BAL are not technically feasible and affordable in Ethiopia [35].

The updated HIV treatment guidelines in Ethiopia recommend co-trimoxazole prophylaxis initiation among all HIV patients who had CD4 <200 cells per µL [71]. Patients with advanced HIV-disease receive drugs such as co-trimoxazole and fluconazole free of charge unless there is a drug stock out. Co-trimoxazole is a broad-spectrum antibiotic used as a prophylaxis for severe immunocompromised patients in order to prevent PCP, toxoplasmosis, malaria and respiratory and enteric infections. 

Ethiopia is not an endemic area for systemic mycosis infections such as histoplasmosis, paracoccidioidomycosis or blastomycosis and non-tuberculous mycobacterial infections may be indistinguishable from pulmonary tuberculosis. This is especially the case in a tuberculosis-endemic area such as Ethiopia. The proportion of smear-positive pulmonary, smear-negative pulmonary and extra pulmonary tuberculosis were 20%, 34% and 46%, respectively [72]. Some of the above described fungal respiratory infections may have been misdiagnosed and treated as smear-negative tuberculosis cases.

We estimated 15,200 cases of CPA in 2017 among patients who recovered from TB. This large case load of affected people should be incorporated into national tuberculosis treatment guidelines in Ethiopia. To diagnose CPA, an ELISA for *Aspergillus* IgG antibodies or a lateral flow assay is needed [73]. Although widely available in developed countries, in RLSs, they are not usually available, as in Ethiopia. As a result, patients are typically treated with months of anti-tuberculosis medication even with no acid-fast bacillus seen [54]. A new rapid point of care, the *Aspergillus* IgG antibody test may facilitate a change in practice [74].

Our estimate of asthma prevalence in adults (2%) is likely to be a significant underestimate as a systematic review found that 16.3% of Ethiopians had had asthma at some point [75] and these higher prevalence rates are consistent with many other African countries [23]. Clearly, epidemiological data on asthma prevalence and fungal sensitization rates in Ethiopia are required.

We estimated 7,051,700 tinea capitis cases (6716 per 100,000) among children aged between 3 and 15 years in Ethiopia annually. Tinea capitis accounted for the highest prevalence of FIs in the country. This could be due to poor individual and environmental sanitation, limited health facilities, the shortage of water supply at school or home for children, poor handwashing behaviour, strong and close contact with animals, highly populated one-room schools and the tropical region with high humidity [76,77,78].

Candidaemia is a leading cause of nosocomial bloodstream infection across the world [79]. The annual mortality of invasive candidiasis in Ethiopia could be over 3600 due to the unavailability of appropriate diagnosis and antifungals. Immunocompromised patients such as those with haematological malignancy and those in ICUs are at high risk of developing candidaemia and *Candida* peritonitis. Data to support our estimate of the number of invasive candidiasis cases are lacking in Ethiopia. For instance, it is difficult to obtain the exact number of ICU beds in Ethiopia. Therefore, we assumed that up to 5% of all hospital beds in Ethiopia serve as ICU beds. From personal communications, we found that only teaching and referral hospitals have ICU services in Ethiopia. In general, we may underestimate the invasive candidiasis, since we did not consider other risk factors such as prolonged antibiotic therapy and low infection prevention practices in the country.

Our study has several limitations. Despite an exhaustive search, we could not obtain enough data, particularly for the incidence of invasive FIs and associated mortality in the country. So, we used data from other neighbouring countries. We also could not obtain data on how many people from Ethiopia used immunosuppresses agents due to organ transplantations and autoimmune diseases. To our knowledge, there is no official or unofficial registry of patients who have had an organ transplant or those who have had surgeryin the country. We assumed that the lack of data about the risk groups for specific FIs might have resulted in the overestimation or underestimation of the number of some invasive FIs. 

In general, our estimates probably have wide confidence limits, due to the modelling style we used and the lack of evidence data for some invasive FI risk groups from Ethiopia. Despite these shortcomings, the burden of FIs in Ethiopia was addressed for the first time. Policy makers and stockholders may use these results for programmes and interventions in order to improve the diagnosis and management of FIs in the country. 

## 5. Conclusions

To our knowledge, this is the first review report on the burden of FIs in Ethiopia. We estimated that fungi could infect approximately 8% of people in Ethiopia every year. However, the actual number of FIs in the country may be higher than our estimates, since we only used HIV infection to determine the immunosuppressed population and some other immunocompromised populations were not considered. Our estimates could be used as basis for future epidemiological studies and national FI surveillance for Ethiopia. In this study, we addressed the neglected fungal diseases in the country, which was a major public concern. Our estimates clearly point to the need for national FI surveillance, improving the mycology laboratory and point-of-care diagnostics, and accessibility to optimum antifungal drugs. The CD4 T-cell count test services should be available for opportunistic FI management and prevention for HIV late presenters. We recommend creating awareness for FIs and supporting mycosis research in the country. Upgrading the diagnosis and management of FIs is urgently needed to reduce their high mortality and morbidity in Ethiopia.

## Figures and Tables

**Figure 1 jof-05-00109-f001:**
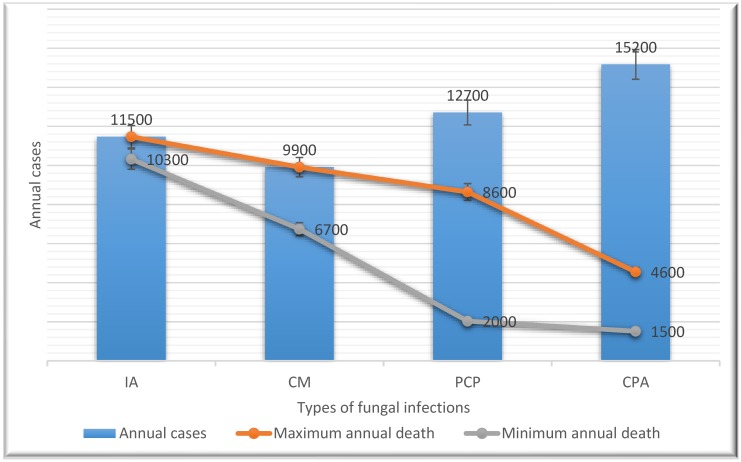
Annual estimates of the number of cases and deaths associated with the most relevant fungal infections in Ethiopia. IA, Invasive aspergillosis; CM, cryptococcal meningitis; PCP, *Pneumocystis jirovecii* pneumonia; CPA, chronic pulmonary aspergillosis.

**Table 1 jof-05-00109-t001:** Country’s profile, populations and rates required to calculate the burden of fungal infections.

**Demographic data**	Total population = 105,000,000 of children (<15 years) = 41% Total number of adults = 60,180,000 Adult women = 31,293,600 % women over 60 = 5% % women 15–50 = 24,482,600	Source: WHO Tuberculosis (TB) stats 2011 [21]; http://aidsinfo.unaids.org/
	Children age 4–14 years = 21,832,000	Source: Education for All 2015 national review report [16]
**HIV/AIDS**	Current total HIV/AIDS in 2017 = 610,000 Proportion of diagnosed cases on ARVs = 440,000 Annual new AIDS cases (at risk of OIs) = 169.5	Source: https://www.unaids.org/en/resources/documents/2018/unaids-data-2018
	Proportion of AIDS patients presenting with *Pneumocystis jirovecii* pneumonia (PCP) = 30%	Source: Aderaye, G. et al. (2008)
	Proportion of AIDS patients presenting with cryptococcal meningitis = 12% = 44,400	Source: Awoke D. et al. (2018)
	AIDS-related deaths in 2017 = 15,000	http://www.who.int/hiv/pub/progressreport2011
**Tuberculosis**	Pulmonary tuberculosis annual incidence Total = 105,200	Source: WHO Global Tuberculosis report 2016 [22]
**Adults with asthma**	Prevalence of asthma in adults = 2.0% Asthma prevalence in adults = 1,239,000	Source: To et al. [23]
**COPD**	COPD prevalence (all GOLD stages) = 7.8% and 10% admitted to hospital annually	Source: Musafiri et al. [24]
**Patients with leukaemia**	AML population = 3000	Source: Globocan (2018)
**Lung cancer**	Estimation of lung cancer = 2000	Source: Globocan (2018)
**All cancer sites**	otal cancer incidence = 67,600	Source: Globocan (2018)
**Number of critical care patients**	Estimated number of critical care beds = 1500 (5% of hospital beds in Ethiopian) Hospital beds (per 1000 people) in Ethiopia = 0.7	Source: World Bank, country Indicator: Number of hospital beds

COPD, chronic obstructive pulmonary disease; GOLD, Global initiative for Obstructive Lung Disease; ARV, antiretroviral; AML, acute myeloid leukaemia.

**Table 2 jof-05-00109-t002:** Incidence and prevalence rates previously reported used to estimate the burden of fungal infections.

Diseases	Population	Prevalence	Reference
**Cryptococcal Infection**	Cryptococcal antigen (CrAg)–positive HIV-infected persons <150 CD4 cells/µL	6.2%	Beyene et al. [25]
	Cryptococcal antigenemia among HIV-infected patients receiving antiretroviral therapy in Ethiopia	8.4%	Alemu et al. [26]
	Cryptococcal disease among hospitalized HIV–infected adults in Ethiopia	9.1%	Mamuye et al. [27]
	Cryptococcal antigenemia ART-naïve and experienced HIV-infected persons	10.2%	Beyene et al. [28]
	Cryptococcal antigenemia among HIV-infected patients at a referral hospital	11.7%	Derbie et al. [29]
**PCP**	Pulmonary infections in TB smear-negative HIV-positive patients with atypical chest X-ray in Ethiopia	29.7%	Aderaye et al. [30]
	HIV-positive patients with suspected pulmonary tuberculosis in Ethiopia	10.9%	Aderaye et al. [31]
**Oral candidiasis**	PLWHA ART-naive with CD4 <200 cells per µL	90%	Fabian et al. [32]
**Oesophageal candidiasis**	AIDS patients in Denmark. In total, 20% of patients with HIV not on ARVs and 5% of those on ARVs	20% ART-naive and 5% of those on ART	Smith and Orholm [33]
**ABPA**	Adult asthmatics	2.5%	Denning et al. [34]
**CPA**	Chronic pulmonary aspergillosis commonly complicates treated pulmonary tuberculosis with residual cavitation	6.5% annually in those with tuberculosis cavity and 0.2% with no cavities, assumed to be 67% of all TB-related CPA	Page et al. [35]
**SAFS**	Severe asthmatics (adults). In total, 10% of asthmatics have severe asthma	33%	Denning et al. [34]
**rVVC**	Adult women	6%	Sobel et al. [36] Foxman et al. [37]
**Tinea capitis**	School-age children	32.3%	Hibstu et al. [38]
	Children	29%	Figueroa et al. [39]
	Children from grade 1 to grade 4	27.4%	Murgia et al. [40]
	Primary school children in rural areas	24.6%	Leiva-Salinas et al. [41]

CrAg, cryptococcal antigen; PLWHA, people living with HIV/AIDS; ART, antiretroviral therapy; ABPA, allergic bronchopulmonary aspergillosis; CPA, chronic pulmonary aspergillosis; SAFS, severe asthma with fungal sensitisation; rVVC, recurrent vulvovaginal candidiasis; IQR, interquartile range; TB, Tuberculosis; PCP, *Pneumocystis jirovecii* pneumonia.

**Table 3 jof-05-00109-t003:** The estimated annual case load of serious fungal infections in Ethiopia.

Fungal Infection	Predominant Groups at Risk	Rate Per 100,000	Estimated Number of Cases
Cryptococcosis	AIDS	9.4	9900
PCP	AIDS	12.1	12,700
IA	Haematological Malignancy, lung cancer, 4% of AIDS deaths and 1.3% of COPD admissions to hospital	10.9	11,500
CPA	Tuberculosis patients and other respiratory disorders	14.5	15,200
ABPA	Adult asthma patients	28.6	30,100 *
SAFS	Adult asthma patients	38.9	40,900 *
Candidaemia	Hospitalised patients	5.0	5300
Candida peritonitis	Post-surgical patients	0.8	800
Oral candidiasis	HIV patients	72.7	76,300
Oesophageal candidiasis	HIV infection	53.2	55,900
Recurrent vaginal candidiasis	Adult women	1399	1,469,100
Tinea capitis	4–14-year-old children with poor hygiene	6716.0	7,051,700
Total burden estimated			8,738,300

ABPA, allergic bronchopulmonary aspergillosis; IA, invasive aspergillosis; SAFS, severe asthma with fungal sensitisation; CPA, chronic pulmonary aspergillosis; COPD, chronic obstructive pulmonary disease. * Duplication between ABPA and SAFS is likely as both are sensitised to *Aspergillus*.

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
