# Peer review of "The Burden of Fungal Infections in Ethiopia"

_jof, 2019, doi:10.3390/jof5040109_

Round 1

Reviewer 1 Report

In this manuscript, Tufa et al. report their estimate of cases of various fungal infections in Ethiopia, including cryptococcal meningitis, Pneumocystis jirovecii pneumonia, tinea-capitis, recurrent candida vaginitis, and oral and oesophageal candidiasis. This estimation was based on current total population and rate of various fungal infections in the last several years in Ethiopia and other neighboring countries. The information presented in the manuscript has very little value both in science and practice. The manuscript is poorly written, especially the discussion section, which simply reiterated everything that has been described in Introduction, Method, and Result sections. There are also some logic, syntax, and grammatical problems.

Author Response

October 25, 2019

Re:  jof-590317-1

Dear Sir/Madam,

We would like to thank you for your concrete comments for our manuscript under consideration for the Journal of Fungi. In reference to a major revision of our manuscript submission, we have revised and integrated the comments to the manuscript to meet the Journal of Fungi criteria for publication.  We include a revised manuscript with typos corrected.

Reviewer #1:

1. In this manuscript, Tufa et al. report their estimate of cases of various fungal infections in Ethiopia, including cryptococcal meningitis, Pneumocystis jirovecii pneumonia, tinea-capitis, recurrent candida vaginitis, and oral and oesophageal candidiasis. This estimation was based on the current total population and rate of various fungal infections in the last several years in Ethiopia and other neighboring countries. The information presented in the manuscript has very little value both in science and practice.

Response: We have tried to use the recently published data for opportunistic fungal infections like CM and PCP since the burden could be influenced by the expansion of ART services. For tinea capitis also we used the recent data.  In general, we have tried to estimates the burden of fungal infections among specific risk groups. This and similar manuscripts do provide a gap analysis of what is and what is not known about the epidemiology of fungal diseases in the country and the many diagnostic deficiencies. We hope the substantial numbers estimated will become a wake-up call to the country’s public health leaders to address the knowledge and care deficits.

The manuscript is poorly written, especially the discussion section, which simply reiterated everything that has been described in the Introduction, Method, and Result sections. There are also some logic, syntax, and grammatical problems.

Response: the grammatical problems and repetition of some sentences were avoided. We have done some further editorial improvements.

In general, we have further edited and improved grammar in the whole paper according to the reviewers’ comments.

Cordially,

Tafese B. Tufa

Arsi University College of Health Sciences,

Asella, Ethiopia.

Phone: +251911771893

Skype: tafeseb.tufa

Reviewer 2 Report

The paper submitted by Tafese Tufa and David Denning provides estimates on the burden of severe fungal diseases in Ethiopia.

Abstract: Informative and complete. Introduction: Concise, providing appropriate background information. Methods: The second part of the second paragraph needs stylistic editing. Results: Demographic data: Please check the numbers for children – they are not congruent. HIV/AIDS: Pl. clarify whether cases of PCP and CM are related to the total number of patients or those with a new diagnosis in the respective year. Other sections: No queries or comments. Discussion: No comments or recommendations. Conclusion: Appropriate with presenting the challenges and the next most important steps to tackle the morbidity imposed by FIs.

Author Response

October 25, 2019

Re:  jof-590317-1

Dear Sir/Madam,

We would like to thank you for your concrete comments for our manuscript under consideration for the Journal of Fungi. In reference to a major revision of our manuscript submission, we have revised and integrated the comments to the manuscript to meet the Journal of Fungi criteria for publication.  We include a revised manuscript with typos corrected.

Reviewer #2:

1. Abstract: Informative and complete. Introduction: Concise, providing appropriate background information. Methods: The second part of the second paragraph needs stylistic editing.

Response: The journal only allows us to use a single paragraph of about 200 words maximum without headings. However, we have tried to follow the abstract format as background, method, result, and conclusion with in a single paragraph. The second paragraph was edited.

Results: Demographic data: Please check the numbers for children – they are not congruent.

Response: Children age less than 4 years are not included in this group (4-14 years =21,832,000 ) due to they do have less risk have a tinea capitis.

HIV/AIDS: Pl. clarify whether cases of PCP and CM are related to the total number of patients or those with a new diagnosis in the respective year.

Response: The cases of PCP and CM are the possible total cases that could be diagnosed as new cases in the respective two years.

Appropriate with presenting the challenges and the next most important steps to tackle the morbidity imposed by FIs.

Response: Fungal infection is one of the most neglected diseases in Ethiopia. There is no systematic review, which can summarize the burden of diseases like other neighbor countries. This why we are interested to estimate the burden of the diseases despite the scarcity of reliable published data in the country.

In general, we have further edited and improved grammar in the whole paper according to the reviewers’ comments.

Cordially,

Tafese B. Tufa

Arsi University College of Health Sciences,

Asella, Ethiopia.

Phone: +251911771893

Skype: tafeseb.tufa

Reviewer 3 Report

This estimate of fungal infections in Ethiopia represents an important first step in characterizing the burden of these infections and associated mortality in this country. The authors readily acknowledge the many limitations inherent in such estimates, not least of which is the absence of statistical modeling or confidence intervals. The authors could consider using more low and high model inputs and scalars to estimate broader estimate ranges that better reflect the uncertainty involved. If this is not done, I would suggest being clearer in the paper, and in the abstract in particular, that these are crude estimates that principally result from use of fixed multipliers used on rough population estimates. As currently written, the numbers convey a sense of precision not supported by the data. That said, I think the current approach provides a reasonable estimate of the general magnitude of these infections.

The paper would benefit from additional proofreading and English-language editing. There are number of missing or excessive spaces. Here are a few examples of problematic syntax: “Google scholars,” “risk groups of FIs have unsimilarity,” “met analysis,” “with develop,” “are the most cause of,” “cryptococcosis meningitis,” “we were enforced.” The discussion contains important and useful points but needs substantial editing for clarity. It also needs reworking to better highlight study findings and context in a useful way, as it is currently quite choppy and hard to follow.

On Figure 1, I’d suggest using the same scale for cases and deaths. On initial read, it looks odd that the number of deaths from IA exceeds the number of cases. The scales are so similar that there seems to be no reason to use different levels.

Regarding the Sobel et al estimate of recurrent vulvovaginal candidiasis, the authors should note that these numbers have been flagged as being inadequately documented in the literature. https://bmcwomenshealth.biomedcentral.com/articles/10.1186/1472-6874-14-43 “Highly-cited estimates of the cumulative incidence and recurrence of vulvovaginal candidiasis are inadequately documented”

Author Response

October 25, 2019

Re:  jof-590317-1

Dear Sir/Madam,

We would like to thank you for your concrete comments for our manuscript under consideration for the Journal of Fungi. In reference to a major revision of our manuscript submission, we have revised and integrated the comments to the manuscript to meet the Journal of Fungi criteria for publication.  We include a revised manuscript with typos corrected.

Reviewer #3:

This estimate of fungal infections in Ethiopia represents an important first step in characterizing the burden of these infections and associated mortality in this country. The authors readily acknowledge the many limitations inherent in such estimates, not least of which is the absence of statistical modeling or confidence intervals. The authors could consider using more low and high model inputs and scalars to estimate broader estimate ranges that had better reflect the uncertainty involved. If this is not done, I would suggest being clearer in the paper, and in the abstract in particular, that these are crude estimates that principally result from the use of fixed multipliers used on rough population estimates. As currently written, the numbers convey a sense of precision not supported by the data. That said, I think the current approach provides a reasonable estimate of the general magnitude of these infections.

Author Response: We have accepted the reviewer comments and we have rounded the numbers to the nearest 100 to be less precise and modified the sentences to reflect the presence of possible uncertainty.

The paper would benefit from additional proofreading and English-language editing.

Author Response: We have accepted the comments and edited the language as well as a little modified the structure of the discussion part.

In Figure 1, I’d suggest using the same scale for cases and deaths. On an initial read, it looks odd that the number of deaths from IA exceeds the number of cases. The scales are so similar that there seems to be no reason to use different levels.

Author Response: Correct. The figure was adjusted.

Regarding the Sobel et al estimate of recurrent vulvovaginal candidiasis, the authors should note that these numbers have been flagged as being inadequately documented in the literature. https://bmcwomenshealth.biomedcentral.com/articles/10.1186/1472-6874-14-43 “Highly-cited estimates of the cumulative incidence and recurrence of vulvovaginal candidiasis are inadequately documented”

Response: The citation was added. You could found on reference number 59.

In general, we have further edited and improved grammar in the whole paper according to the reviewers’ comments.

Cordially,

Tafese B. Tufa

Arsi University College of Health Sciences,

Asella, Ethiopia.

Phone: +251911771893

Skype: tafeseb.tufa

Round 2

Reviewer 1 Report

There is little improvement in this manuscript. Other than some wording changes, the content remains the same.